# Genetic Mapping of Genotype-by-Ploidy Effects in *Arabidopsis thaliana*

**DOI:** 10.3390/genes14061161

**Published:** 2023-05-26

**Authors:** Cris L. Wijnen, Frank F. M. Becker, Andries A. Okkersen, C. Bastiaan de Snoo, Martin P. Boer, Fred A. van Eeuwijk, Erik Wijnker, Joost J. B. Keurentjes

**Affiliations:** 1Laboratory of Genetics, Wageningen University & Research, Droevendaalsesteeg 1, 6708 PB Wageningen, The Netherlands; 2Biometris, Wageningen University & Research, Droevendaalsesteeg 1, 6708 PB Wageningen, The Netherlands; 3Rijk Zwaan R&D Fijnaart, Eerste Kruisweg 9, 4793 RS Fijnaart, The Netherlands

**Keywords:** *Arabidopsis thaliana*, QTL mapping, ploidy, flowering

## Abstract

Plants can express different phenotypic responses following polyploidization, but ploidy-dependent phenotypic variation has so far not been assigned to specific genetic factors. To map such effects, segregating populations at different ploidy levels are required. The availability of an efficient haploid inducer line in *Arabidopsis thaliana* allows for the rapid development of large populations of segregating haploid offspring. Because Arabidopsis haploids can be self-fertilised to give rise to homozygous doubled haploids, the same genotypes can be phenotyped at both the haploid and diploid ploidy level. Here, we compared the phenotypes of recombinant haploid and diploid offspring derived from a cross between two late flowering accessions to map genotype × ploidy (G × P) interactions. Ploidy-specific quantitative trait loci (QTLs) were detected at both ploidy levels. This implies that mapping power will increase when phenotypic measurements of monoploids are included in QTL analyses. A multi-trait analysis further revealed pleiotropic effects for a number of the ploidy-specific QTLs as well as opposite effects at different ploidy levels for general QTLs. Taken together, we provide evidence of genetic variation between different Arabidopsis accessions being causal for dissimilarities in phenotypic responses to altered ploidy levels, revealing a G × P effect. Additionally, by investigating a population derived from late flowering accessions, we revealed a major vernalisation-specific QTL for variation in flowering time, countering the historical bias of research in early flowering accessions.

## 1. Introduction

Although common in some species, phenotypic effects caused by differences at genome ploidy level have so far been very elusive and difficult to study in the plant model *A. thaliana*. Nonetheless, the impact of ploidy is illustrated by strong effects on quantitative traits such as salt and drought tolerance, and relative growth rate [1,2,3]. Most attempts to reveal ploidy effects in Arabidopsis have used naturally occurring autotetraploid accessions such as Warschau-1 (Wa-1) [4,5] or artificially induced tetraploids [6], which were compared to their diploid and triploid counterparts [3,4]. However, to identify genetic factors that are causal for the observed differences in response to altered ploidy levels, segregating populations are required. While a biparental mapping population was developed using Wa-1 as one of the parental genotypes, it was only later discovered that this genotype was tetraploid and that the inbred lines were derived from triploids [4,7]. Therefore, the ploidy level segregated in this population and many of the genotypes are not explicitly diploid or tetraploid according to flowcytometry [1,4]. Notwithstanding this unstable population, a mapping resource at different stable ploidy levels has not been developed systematically in Arabidopsis so far.

Monoploids (i.e., individuals consisting of somatic cells containing only the basic number of chromosomes) are usually not taken into account in studies that investigate the effect of ploidy, although exceptions exist in maize [8], yeast [9], potato [10], and Chinese cabbage [11]. These studies focused on transcriptional changes induced by ploidy changes in a single or a few genotypes. For instance, Stupar et al. [10] demonstrated that more than 50% of the analysed genes displayed expression differences between monoploids and diploids or tetraploids, suggesting large developmental differences between plants of different ploidy levels. The discovery of a genome elimination mutant in Arabidopsis allowed the quick generation of haploid lines from diploid individuals [12], enabling the analysis of ploidy effects in a genetic model species. The generation of haploids in Arabidopsis occurs through elimination of the mutant haploid inducer genome in the offspring of a cross between the mutant and a wild type diploid, and can easily be distinguished from aneuploids and diploids. Diploid Arabidopsis somatic cells contain 2n = 2x = 10 chromosomes, while haploids contain n = x = 5 chromosomes and thus are equivalent to monoploids. Arabidopsis monoploids are predominantly sterile and cannot be maintained as such. Haploid plants do, however, occasionally set seed, giving rise to homozygous doubled haploids (DHs). Spontaneous diploidisation occurs during sexual reproduction due to fusion of euploid gametes resulting from incidental non-disjunction of all homologs at meiosis I or through somatic doubling of haploid cell lines, resulting in chimeric plants with fertile diploid branches. While the ploidy level of the maternally derived seed coat is determined by that of the mother plant, the embryo and endosperm (2n = 2x and n = 3x*,* respectively) contain equal chromosome numbers in seeds derived from mono- or diploids. Thus, doubled haploids contain a duplicated genome that consists of diploid somatic cells containing 2n = 2x = 10 chromosomes, identical to wild type diploids.

When the genome elimination mutant is crossed with an F_1_ hybrid of two distinct accessions, only the recombinant gametes of the hybrid will contribute to the genomes of the resulting monoploid offspring. By allowing the monoploids to produce DH seeds, the monoploid genome is immortalized in homozygous diploids, resembling recombinant inbred lines (RILs). The generation of such a diploid mapping population using genome elimination thus has the advantage that initially large amounts of segregating monoploids are produced, which except for the ploidy level, are genetically identical to the DHs obtained in the next generation [13,14,15]. These monoploids may provide a useful additional resource for genetic mapping and allow assessment of ploidy effects in comparisons with their subsequent isogenic diploid offspring.

The generation of DH mapping populations has an advantage over the more commonly used RILs, which are typically generated from an F_1_ individual through eight to ten generations of self-fertilisation. This contrasts DHs for which homozygous diploid populations can be obtained from an F_1_ in only three generations, although the resolution of DH populations might be lower due to a reduced number of recombination events [13,16,17,18]. The advantage of the fast development of DH populations allows for the investigation of natural variation in late flowering winter annual accessions, whereas most existing experimental biparental mapping populations are derived from summer annual accessions to shorten the generation time due to their early flowering phenotype [17,19,20,21,22]. Summer annuals germinate in spring and flower within a short period of time, while winter annuals germinate in autumn, survive winter as a rosette and typically flower only after vernalisation, a period of cold conditions. These differences may have a large impact on life history traits, but despite an increase in genetic resources, including more accessions to represent the huge global genetic diversity of the species, a bias towards the use of early flowering accessions remains [20,21,22]. Illustratively, although the haploid inducer approach eliminates the need for a lengthy inbreeding process to obtain homozygous lines, the DH populations reported for Arabidopsis so far also originate from early flowering accessions [13,15,23,24].

Here, we describe the development and phenotyping of a monoploid, and subsequent diploid mapping population derived from a cross between the two late flowering accessions, T540 (Kävlinge, Sweden) and Ge-0 (Geneva, Switzerland). These accessions display large phenotypic differences in a number of life history traits. We investigated the diploid generation for the presence of a genotype-by-environment (G × E) effect by mapping variation in flowering time with and without vernalisation. We demonstrate that exploiting genetic variation in late flowering accessions can increase our knowledge even in a well-studied trait like flowering time. Secondly, we investigated the possibility of detecting genotype-by-ploidy (G × P) interactions by performing a combined analysis across monoploids and diploids, using a multi-trait QTL model approach. As such, we were able to detect ploidy-specific QTLs and reveal genotype-by-ploidy interactions. Finally, we analysed all traits for pleiotropic QTLs, and demonstrated that most detected QTLs affect multiple traits at both ploidy levels, while only a minor number of QTLs affect predominantly a single trait at a specific ploidy level. Taken together, this study advocates for the use of late flowering mapping populations to analyse as of yet unexploited genetic variation and provide evidence for genotype-by-ploidy interactions in Arabidopsis.

## 2. Results

### 2.1. Development and Phenotyping of a Mapping Population at Two Ploidy Levels

To explore the effect of ploidy on genetic mapping in Arabidopsis, a segregating population was generated from a cross between two late flowering accessions, T540 and Ge-0 (Figure 1). Briefly, the late flowering accessions T540 and Ge-0 were crossed to produce an F_1_ hybrid. This hybrid was subsequently manually crossed to a haploid inducer line [12], from which approximately 250 seeds were obtained. These seeds were stratified and pre-germinated, after which seedlings were transferred to Rockwool and grown for three weeks under long day conditions in a climate-controlled growth chamber. After visual inspection, 210 potentially haploid plants were transferred to a cold room for eight weeks vernalisation under short day-length conditions. Once vernalised, plants were transferred to a greenhouse under long day conditions and subsequently formed inflorescences, flowered, and set seeds. At the end of the growth period, non-destructive phenotypes were measured, i.e., main stem length, branching from rosette, and branching from the main inflorescence (Appendix A), allowing the monoploids to produce doubled haploid seeds. These seeds formed the subsequent diploid generation. The diploid seeds harvested from monoploid plants were also analysed for average seed size (Appendix A).

In a second experiment, the 210 potential DH lines were grown in a climate chamber under similar conditions as described for the monoploids. Ten replicates of each line of the diploid population were grown in a completely randomised design. After three weeks, five of these were transferred to a greenhouse to record the time to flowering. The other five replicates were transferred to a cold room and vernalised for eight weeks at 4 °C. These plants were thereafter transferred back to the climate chamber with long day conditions and phenotyped for flowering time in addition to the life history traits quantified in the monoploids (Appendix A). Assuming all replicate plants were isogenic, one plant of each genotype was selected for genotyping, which was successful for 195 lines (Appendix A). After analysis of genotypic and phenotypic data, 171 genotypes, for which phenotypic data at both ploidy levels could be obtained, were selected. The phenotypic data of these lines were used for all further analyses. The genotype data of these lines were used for the construction of a genetic map (Appendix A) and the QTL mapping of the analysed traits using standard methods.

In addition to the artificial haploid and DH mapping populations, a classical F_2_ population of 400 lines derived from the same T540 × Ge-0 F_1_ hybrid was generated and grown simultaneously with the doubled haploids in the second experiment. Half the population was subjected to vernalisation again, while the other half was left to flower (Appendix A). Moreover, a small set of 71 vernalised F_2_s was genotyped with the same 123 markers as used to genotype the DHs and their linkage patterns were compared with those in the DH population to confirm no anomalies occurred during the DH development (Appendix A). With the exception of a slight genotype distortion at the top of chromosome 1 in the DH population (Appendix A), no systematic differences were observed between the F_2_ and the DH population. Importantly, the genetic maps generated from the two populations displayed an almost identical marker order, consistent with the known physical position of markers (Appendix A).

### 2.2. Detection of Genetic Variation Controlling Flowering Time Conditional on Vernalisation

While a vernalisation treatment can have a large overall phenotypic effect on the morphology and inflorescence structures of late flowering accessions [25,26], mapping experiments in Arabidopsis have focused on detecting QTLs in either early flowering populations or on mapping specific QTLs involved in vernalisation requirement with populations derived from parental accessions differing in this aspect [19,20,21,22,27]. Here, we have the opportunity to compare and map natural variation in flowering time with and without vernalisation in a late flowering segregating DH and F_2_ population.

Although both parents of the DH and F_2_ population are late flowering, they do not per se require vernalisation to flower. In our experiment without vernalisation, T540 flowered on average after 101.8 days after sowing (DAS), while this was 86.5 days for Ge-0 (Table 1 and Appendix A). With vernalisation, these accessions flowered on average after 19.1 and 14.2 days after transfer (DAT) from the cold, respectively. Similar data were obtained for the F_2_ population (Table 1 and Appendix A). The variation in flowering time between the two accessions segregated in the diploid populations with only minor transgression in both conditions. Without vernalisation, the earliest line of the DH population flowered after 63 DAS, while the latest flowered at 123 DAS. With vernalisation, the difference between extreme lines reduced to only ten days (12 and 22 DAT, respectively). The correlation in flowering time between the vernalised and non-vernalised plants was positive but far from absolute (R^2^ = 0.39) (Appendix A).

The data for flowering time of the doubled haploids under different vernalisation conditions allowed for a multi-environment composite interval mapping (CIM) where the effect of vernalisation was investigated. Additionally, the F_2_ population was screened for QTLs in vernalised conditions in a separate analysis. A total of seven QTLs spread over the genome were detected for variation segregating in the DH population (Table 2). Of these seven QTLs, three revealed an interaction with the environment, providing evidence for G × E effects of vernalisation. One QTL with a G × E effect was located on chromosome 4 and the other two were detected on chromosome 5. The QTL for flowering time after vernalisation in the middle of chromosome 4 had a normalized effect-size of 0.33 (indicating a positive contribution of the T540 allele), while this QTL was not significantly (*p* = 0.653) detected in the non-vernalised DH population. In contrast, both G × E QTLs on chromosome 5 were significant in both environments but with different effect-sizes (Table 2). The major QTL detected at the bottom of chromosome 5 revealed an additive normalized effect-size of 0.65 when the plants were vernalised, while this was only 0.19 in the non-vernalised set. This major QTL was also detected in the vernalised F_2_ population. The other G × E QTL detected in the middle of chromosome 5 in the DH population had a normalized effect-size of 0.43 in non-vernalised plants, while this was substantially lower (0.21) in vernalised plants. Additional QTLs without G × E effects were detected in the middle and at the bottom of chromosome 1 and chromosome 2 and at the top of chromosome 3 (Table 2). These results indicate that, conditional on the environment, genetic variation can have a variable impact on the time to flower in Arabidopsis.

### 2.3. Effects of Ploidy Level on Morphological Variation

To investigate if differences in ploidy level have an effect on the phenotype, various morphological traits were quantified in the mono- and diploid generation of the recombinant lines generated from the cross T540 × Ge-0 (Figure 2 and Appendix A). Monoploid recombinant lines were on average much taller than their diploid counterparts (65 versus 46 cm, respectively) (Figure 2A and Appendix A). Illustrative for this difference in length is that more than 60% of the monoploids grew taller than the tallest diploid, which measured only 61 cm. In addition, branching from the rosette occurred much more frequently in monoploids (95.8%) than in diploids (34.9%), resulting in a larger average number of branches sprouting from the rosette and, similar to main stem length, more pronounced variation (Figure 2B; Appendix A). Illustratively, a maximum of only three rosette branches was observed in diploids, while monoploids developed on average seven branches from the rosette, with an exceptional maximum of twenty-three branches. In contrast to variation in main stem length and branching from rosette, the variation in branching from the stem spread around almost identical mean values at both ploidy levels, although a larger transgression was observed in the monoploids as compared to the diploids (Figure 2C). Despite the differences in phenotypic variation between the number of branches from the rosette and from the stem, these traits correlated positively in the monoploid population (R^2^ = 0.32) (Appendix A). This resulted in monoploids with up to a total number of thirty-two branches, giving rise to a bushy phenotype.

Similar to branching from the stem, the phenotypic variation in the size of seeds harvested from mono- or diploid plants centered around a comparable mean for both types of population, although the between-line variation was somewhat larger for seeds derived from diploids than for those derived from monoploids (Figure 2D and Appendix A). Positive Pearson correlations between mono- and diploids were observed for all traits (Appendix A), but values remained moderate (0.3 < R^2^ < 0.4). The replicate measurements of the diploid genotypes also allowed the assessment of trait heritabilities (Appendix A). For most traits, segregating in the diploid population moderate to high broad-sense heritabilities were obtained (0.30 < H^2^ < 0.83). This suggests that differences between mono- and diploids can be partly explained by simple additive ploidy effects but that the larger part of variation might be the result of more complex genotype-by-ploidy interactions.

### 2.4. Effects of Genotype-by-Ploidy Interaction on the Detection of QTLs

To determine whether differences in ploidy level had an effect on mappable genetic variation, each of the four traits measured in both the mono- and diploid population were subjected to trait-specific dual-trait CIM, in which measurements at the two ploidy levels were considered to be different traits. Significant QTLs could be detected for each trait in both generations. In total, fifteen QTLs were detected for the various traits, of which six displayed a significant interaction with the ploidy level (Table 3). Three genotype-by-ploidy QTLs were detected for main stem length, while one G × P QTL was detected for each of the other traits.

For main stem length, five QTLs were detected in total, with a major QTL on the top of chromosome 5 and minor QTLs on chromosomes 3 and 4 (Table 3). The Ge-0 allele at the major QTL at chromosome 5 increased the stem length in the monoploids (normalized effect-size 0.49), whereas genotypic variation at this locus had no significant influence on the length of the diploids (effect-size 0.03). The QTL on chromosome 3 showed a similar pattern with a significant genotype-effect in the monoploids, although with smaller effect-size than the QTL on chromosome 5, but not in the diploids. Finally, on chromosome 4, three QTLs with overlapping support intervals spanning the entire chromosome and similar effect-signs were detected.

For both variation in branching from the rosette and branching from the main stem, three QTLs were detected (Table 3). For variation in the number of branches from the rosette, QTLs were detected on the bottom of chromosomes 3 and 5 and the top of chromosome 5. The QTL on the bottom of chromosome 5 revealed a clear G × P interaction, as it was highly significant in the monoploids (*p* < 0.001) while it was not detected in the diploid generation (*p* = 0.642). The Ge-0 genotype at this QTL explained an increase in the number of branches in the monoploids, while a Ge-0 genotype at the two other QTLs decreased the number of branches from the rosette at both ploidy levels. Another G × P QTL was detected for variation in branching from the main stem on the middle of chromosome 5. This QTL was significantly detected in the diploids (*p* < 0.001) but not in the monoploids (*p* = 0.204). Similar to an increase in main stem length, Ge-0 alleles at any of these three QTLs increase the number of branches.

Finally, four QTLs were detected for variation in seed area, of which a G × P interaction was identified for the QTL on the middle of chromosome 3 (Table 3). This QTL was significantly detected in the monoploids (*p* = 0.002) but not in the diploids (*p* = 0.224). However, this QTL exerted only a minor effect. Another QTL on chromosome 3 was significantly detected in both generations, although it was much weaker in the diploids (*p* = 0.043) and a large difference in the effect-size of the QTL was observed (0.42 and 0.17 for monoploids and diploids, respectively). The results of the dual ploidy QTL analysis clearly indicate that differences in ploidy do not affect every genotype and trait equally. Indeed, strong G × P QTLs explain for a large part the phenotypic differences observed between genotypes and ploidy levels.

### 2.5. Pleiotropic Effects of Genotype-by-Ploidy Interactions

A weak to moderate correlation could be observed between values of the different morphological traits measured in the two isogenic populations of different ploidy (Appendix A). These relationships suggest a partial co-regulation of traits. Indeed, we detected QTLs at similar positions for multiple traits (Table 3). We, therefore, subjected the various traits measured in the monoploids and diploids after vernalisation to a single multi-trait CIM analysis to identify possible co-location of QTLs. A total of nine QTLs were detected using this approach (Figure 3 and Appendix A). None of these QTLs were trait-specific and only the minor QTLs on the bottom of chromosomes 1 and 4 were ploidy-specific (*p* < 0.01), although suggestive QTLs (*p* < 0.05) were detected for other traits or at the other ploidy level as well (Appendix A).

A minor QTL on the lower arm of chromosome 3 significantly (*p* < 0.01) explained variation in all monoploid traits, but only in branching from the rosette in the diploids. The T540 allele at this locus increases the number of branches from the rosette in the monoploids and diploids, even though the diploids did not display a large variation for this trait. Additionally, the same T540 allele causes an increase in branches from the stem in monoploids. However, the same allele decreases main stem length and seed size of the monoploids. Additional minor to moderate QTLs co-locating on the lower arm of chromosomes 2 and 4 and in the middle of chromosome 3 were detected, explaining variation in multiple traits in both the mono- and diploids. The sign and effect-size of these coinciding QTLs was in line with the observed correlation between these traits (Appendix A).

By far, the strongest and largest number of QTLs was detected on chromosome 5. Strong QTLs for variation in main stem length and rosette branching in the monoploids coincided at the top of the chromosome, although with opposite effect-sign (Appendix A). Another strong QTL for variation in the size of seeds derived from monoploids at 61.2 cM coincided with highly significant QTLs for variation in stem branching, flowering time after vernalisation, and main stem length of diploids. Finally, close to the end of the chromosome (121.7 cM), a strong QTL for variation in main stem length and branching of the monoploids co-located with a QTL for variation in flowering time after vernalisation and branching from the rosette of diploids. The Ge-0 allele at this locus increased all trait values except flowering time after vernalisation, which was delayed by the T540 allele.

Since genetic variation at the two QTLs at the top and bottom of chromosome 5 has the strongest effect on branching and main stem length (in addition to flowering time in the diploids), we analysed the effect of each of the four possible haplotypes in both the monoploid and diploid populations. Reflecting the absence of a significantly detected QTL for variation in stem length and branching at the top of chromosome 5 in the diploids, genotypic variation at the two QTLs had a much stronger effect on the monoploids (Appendix A). This clearly indicates that the effect of genetic variation can be much stronger in monoploids than in diploids (Figure 2).

## 3. Discussion

### 3.1. Application of a Late Flowering Doubled Haploid Mapping Population

It is well known that different accessions of Arabidopsis respond differently to environmental conditions [26,28]. For instance, day-length sensitivity and vernalisation requirement determine for a large part the discrimination between winter and summer annuals [29,30]. Moreover, when mapping populations are subjected to short or long day-length conditions with or without vernalisation, differences in the number and strength of detected flowering time QTLs can be observed [31]. The use of a haploid inducer line in this study allowed the generation of a homozygous mapping population from underexploited late flowering accessions. As such, a diploid population could be developed in only three generations. We acknowledge that flowcytometry might have unambiguously confirmed the ploidy level of the generated resources, but genome elimination has been extensively investigated cytogenetically [12] and has been proven to be a reliable method to generate haploids and doubled haploids in many publications following this initial study [14]. Moreover, haploid plants are unique and cannot be maintained or replicated. Taking tissue samples for ploidy measurements might disturb plant development too much to acquire accurate phenotypic measurements. We, therefore, chose to prioritize on the phenotyping and rely on other assessments for ploidy and genotyping. Furthermore, haploid plants can easily be distinguished morphologically from aneuploids and diploids in an early developmental stage. In addition, haploid plants are sterile while diploids are not and doubled haploids can only result from haploids in contrast to aneuploids (Appendix A). Finally, alternative genotypes (e.g., diploids resulting from failing genome elimination) would be identified by genotyping and including such lines would dramatically have confounded genetic map construction and QTL mapping. We have not experienced either of these phenomena. For those reasons, we are quite confident that our assumptions about the ploidy level of the investigated resources are correct and we, therefore, saw no reason to validate this with flow cytometry.

For the established DH population, QTL mapping for variation in flowering time in two different environments (i.e., with and without vernalisation) was performed. In addition to a number of minor QTLs, a major QTL for variation in flowering time of vernalised plants was detected near the previously described and identified *VERNALISATION INSENSITIVE 3* (*VIN3*; At5g57830) locus at the bottom of chromosome 5 [31,32,33]. Previously, variation in flowering time associated with this locus was explained by an indel of three nucleotides within an exon of *VIN3* [33]. However, this indel is not polymorphic for Ge-0 and T540, although multiple other single nucleotide polymorphisms (SNPs) differentiate the intronic and promotor region of *VIN3* of these accessions, including 28 nucleotides deleted from the T540 *VIN3* promotor compared to Ge-0 (Appendix A).

A second gene, *REDUCED VERNALISATION RESPONSE 2* (*VRN2*; At4g16845), related to response to vernalisation [34], is located within the support interval of a QTL for variation in flowering time after vernalisation, detected on chromosome 4. The VRN2 protein mediates vernalisation through interaction with the Polycomb Group (PcG) protein complex including VIN3 [35,36]. This PcG complex is known to interact with, and cause the stable reduction of the expression levels of, the floral repressor *FLOWERING LOCUS C* (*FLC*; At5g10140) [35,36], which collocates with the position of a flowering time QTL on the top of chromosome 5. This QTL was also detected for variation in main stem length, which strongly suggests a pleiotropic effect on the inflorescence architecture and flowering pathways, previously attributed to *FLC* [25].

The detection of flowering time QTLs in a segregating mapping population of late flowering accessions, especially after vernalisation, clearly identifies major QTLs other than those usually associated with flowering time variation in early accessions. This suggests that the regulation of flowering time in late accessions is controlled by variation at other loci than those in early flowering accessions (e.g., *FRIGIDA* (*FRI*) and *FLC*). It is likely that flowering time is not the only trait that discriminates summer annuals from winter annuals, which advocates for the analysis of traits in late flowering populations in addition to the abundantly available early flowering populations.

### 3.2. Effects of Haploidisation on Phenotypic Variation

Exploiting the availability of a mono- and diploid Arabidopsis mapping population, QTL analyses were applied to map and compare possible ploidy-dependent effects. A dual-trait CIM analysis resulted in the detection of six QTLs with a G × P interaction, while additional QTLs showed large differences in effect-sizes at either ploidy level. An obvious explanation for the G × P QTLs is that monoploid plants are sterile due to unbalanced segregation of the chromosomes during meiosis. Indeed, although not explicitly quantified, monoploids displayed an extended period of flowering compared to fertile diploids, possibly causing the increase in main stem length. Similarly, the development of exceptionally high numbers of rosette branches increases the total number of flowers produced [37]. This suggests that the plants attempt to compensate for the lack of viable seed production by an increase in reproductive tissue formation, implying that the QTLs detected specifically for monoploids might be involved in the response to sterility. A similar phenomenon of additional branch formation has been described for the male sterile Landsberg *erecta* mutant (*ms1-*L*er*) [38]. Nonetheless, QTLs explaining the observed variation in response to haploidisation were detected, indicating natural variation for the strength of ploidy effects.

The antagonistic effect of the QTL on the top of chromosome 5 for either additional rosette branch formation (inferred by the T540 allele) or taller growth (inferred by the Ge-0 allele) implies that both accessions follow a different morphological approach to achieve a similar increase in the number of flowers. The fact that a single QTL is identified for variation in rosette branching and main stem length might be due to one of the many pleiotropic genes that function in the control of inflorescence architecture [39]. Possible candidate genes may be part of the florigen gene family [40] which is known to function as a mobile flowering time switch. For instance, *FLOWERING LOCUS T* (*FT*; At1g65480) and *TWIN SISTER OF FT* (*TSF*; AT4G20370), are known to function in both flower induction and shoot branching pathways [41]. Another member of the same gene family, *TERMINAL FLOWERING 1* (*TFL1*; At5g03840), is located within the support interval of the QTL at the top of chromosome 5 and has been shown to be involved in flowering architecture [42]. Although no variation within the *TFL1* coding sequence could be observed between the two accessions, several SNPs and possibly deletions within the promotor region of the T540 allele might cause a differential expression of this gene (Appendix A). Assuming that flowering architecture is not influenced by VIN3, an alternative candidate explaining the effect of the QTL at the bottom of chromosome 5 on both branching and main stem length is *AUXIN RESPONSE FACTOR2* (*ARF2*; At5g62000), which is involved in multiple developmental processes via cell proliferation [43,44]. Sequence-based evidence suggests that T540 and Ge-0 possess functionally different alleles (Appendix A). Moreover, a knockdown of *ARF2* leads to an increase in stem length and a sterile phenotype [43].

Other QTLs, such as the one explaining variation in the size of seeds on the top of chromosome 3, coincide with likely candidate genes as well. This QTL has been identified previously as *HAIKU* 2 and was associated to a gene (*IKU2*; At3g19700) in the endosperm growth pathway [45]. In addition, a monoploid specific QTL on chromosome 2 explaining variation in branching from the stem coincides with the previously identified *AGAMOUS-LIKE 6* gene (*AG6*; a.k.a. *REDUCED SHOOT BRANCHING 1*; AT2G45650), to which pleiotropic phenotypic effects on both the flowering and branching pathways have been previously attributed [46].

Although sterility might be causal for some of the G × P interactions of the QTLs, it is possible that other molecular processes are of influence as well. In previous studies on ploidy series including monoploids, performed in maize [8], yeast [9], potato [10], and Chinese cabbage [11], differentially expressed genes were identified at different ploidy levels, indicating a specific sensitivity to ploidy, instead of sterility. Moreover, in a dosage series (*x*, 2*x*, 4*x*) of maize inbred lines [47], genetic background and ploidy was suggested to interact. Further evidence for G × P interactions independent of sterility come from an RNA-seq comparison of diploid and tetraploid Arabidopsis accessions, in which the accessions Col-0 and L*er*-0 displayed different numbers of upregulated genes at the tetraploid level [48]. In both studies, it was argued that the altered nuclear surface to volume ratio might have caused the differential expression of genes. However, clear mechanisms explaining how these altered ratio’s cause gene expression differences are so far elusive. Despite the uncertainty of the possible mechanisms of G × P interactions, it is clear that the mapping of quantitative traits in mono- and diploids can reveal additional variation, which might be instrumental in the elucidation of the genetic regulation of complex traits.

## 4. Experimental Procedures

### 4.1. Population Development

Two late flowering accessions, T540 (CS76239) from Sweden and Ge-0 (CS76135) from Switzerland were selected based on phenotypic differences and expected unexplored genotypic differences compared to widely used early flowering accessions. These accessions were crossed to produce a biparental hybrid F_1_. The F_1_ (T540 × Ge-0) was used as a pollen donor and crossed to the GFP-tailswap haploid inducer line to generate monoploid offspring [12]. From these crosses, 250 viable seeds were sown and 210 putative monoploid lines were selected based on morphology during growth [14]. Spontaneous genome doubling in the monoploids followed by selfing created a set of 171 unique diploid homozygous lines. In addition to the generation of the doubled haploid lines, the F_1_ was selfed to generate a batch of F_2_ seeds.

### 4.2. Plant Growth Conditions

All seeds from a cross between the F_1_ hybrid (T540 × Ge-0) and the GFP-tailswap line were sown on ½ MS agar plates without sucrose. The seeds on these plates were stratified for four days at 4 °C in darkness and subsequently placed in a climate chamber at 25 °C with a diurnal cycle of 16 h of light and 8 h of darkness to induce seed germination. After two days of pre-germination, only potential monoploid seedlings were transplanted to wet Rockwool blocks of 4 × 4 cm in a climate chamber (16 h LD, 125 µmolm^−2^s^−1^, 70% RH, 20/18 °C day/night cycle). All plants were watered three days per week for 5 min with 1/1000 Hyponex solution (Hyponex, Osaka, Japan) using flooding tables. Here they remained for three weeks to allow growth before vernalisation. Vernalisation was performed for eight weeks (12 h LD, 125 µmolm^−2^s^−1^, 70% RH, 4 °C constant). After vernalisation, plants were transferred to the greenhouse where they were allowed to flower and mature. Monoploid plants were selected based on morphology as described before [14]. Subsequently, diploid seeds were harvested after recording phenotypic traits of the monoploids.

The second experiment included ten replicates for each of 210 assumed diploids. These were stratified on wet filter paper in similar conditions as the agar plates of the previous experiment. Subsequently, five of the seedlings were grown similar to the monoploids, including three weeks growth in long day conditions and vernalisation for eight weeks, while the five other replicates were transferred to the greenhouse. The five replicates in the greenhouse conditions were allowed to grow in a completely randomized design without vernalisation for a maximum of 100 days after transfer or until flowering or senescence. The five diploids that underwent vernalisation remained in climate chambers with similar conditions as pre-vernalisation (16 h LD, 125 µmolm^−2^s^−1^, 70% RH, 20/18 °C day/night cycle). The plants were randomized in a completely randomized design where they were allowed to grow for a maximum of ninety days.

Similarly, 400 F_2_ plants were grown, of which 200 were vernalised as described above. Of the vernalised plants 71 F_2_s were selected for genotyping.

### 4.3. Phenotypic Measurements

The monoploids were phenotyped for the number of branches from the rosette and branching from the stem, main stem length (cm) and seed area (approximately 100 seeds were taken three times from the same storage bag for three separate photos, these were analysed for seed area). For the second experiment, the same four phenotypes were measured. However, now also flowering time before and after vernalisation was included as a phenotype. Flowering time without vernalisation was measured as the number of days after planting until the first flower on the main stem opened its petals. Flowering time with vernalisation was measured as the number of days after vernalisation until the first flower on the main stem opened its petals. Plants that did not germinate or that died within the period of the experiment were discarded. For the plants used for genotyping only flowering time was recorded, as taking a flower head, used for extracting DNA, from the plant might influence the other traits. All the monoploid and F_2_ phenotypes are based on a single observation per genotype, while for the DH population, which were measured with five replicates, the reported values are the means.

### 4.4. Genotyping of the Populations

For 210 doubled haploids and 71 F_2_s the DNA was extracted from flower heads by applying a CTAB DNA extraction protocol which was adapted for use on 96 well plates. Genotyping was performed using a GoldenGate Assay from Illumina (San Diego, CA, USA), using 384 SNP markers. Of those, 142 markers were polymorphic for the two parental lines. Of these 142, only 114 markers showed nonredundant recombination patterns for either the diploids or F_2_s. Nine additional KASPar markers (KBiosciences, Hoddesdon, UK) were included to a total of 123 markers [49]. From 210 selected DH lines, 195 were successfully genotyped and only four were discarded because of too much heterozygosity or missing data. Eventually, only 171 DHs were used for the final analyses because of redundant genotypes and lack of data in either mono- or diploid generation.

### 4.5. Genetic Map Comparison of the Doubled Haploid and F_2_ Population

To confirm no anomalies were present in the doubled haploids, a comparison with an F_2_ population was performed. Individual lines from both populations were genotyped and genetic maps were generated. A subset of 71 F_2_s and 171 DHs were successfully genotyped. Genetic maps were constructed for both the F_2_ and the DHs independently using Kosambi’s regression mapping function in JoinMap 6.1 (Kyazma, Wageningen, The Netherlands). Segregation distortions were determined by GenStat 19th edition. The DH map was also used for the genetic mapping in monoploids.

### 4.6. Statistical Analyses and QTL Mapping

Pearson correlations between traits were calculated using the cor function in R (R 3.6.0, Vienna, Austria). The broad-sense heritabilities of the doubled haploids were calculated in R using the repeatability function of the heritability package [50]. QTL analyses were performed using GenStat (19th edition) [51], where mean phenotypic values per DH line were used and single observations in the case of the monoploids and F_2_s. In order to have a maximum QTL effect and QTL × E or QTL × Ploidy interaction detection, we first analysed the separate traits using single-trait multiple environment composite interval mapping (where either vernalisation or the ploidy level was considered as the environment). The final analyses encompassed a multi-trait single environment analysis, including all traits measured after vernalisation. First, an initial analysis of simple interval mapping was performed with a maximum step size of 5 cM along the genome. Other settings were kept as default (maximum cofactor proximity = 50 cM; minimum distance for QTL selection = 30 cM; threshold for genome-wide significance level = α = 0.05). After these first analyses, markers associated with candidate QTLs were automatically set as cofactors for the composite interval mapping. The QTLs that resulted from this scan were tested for interaction effects in the selection of a final QTL model.

## Figures and Tables

**Figure 1 genes-14-01161-f001:**
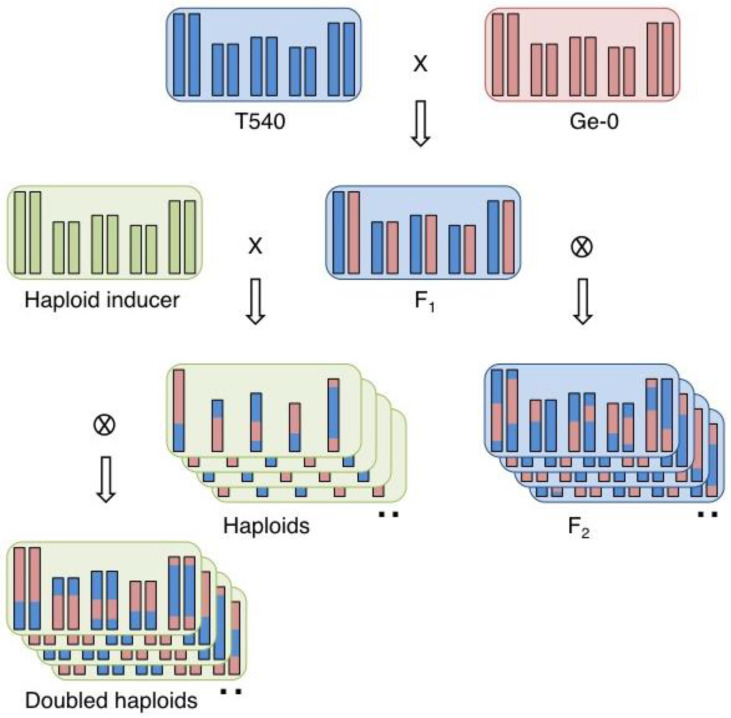
Crossing scheme for the development of haploid and doubled haploid recombinant populations. Each parental genotype (T540 blue; Ge-0 red) is depicted by five double vertical bars, which represent the five chromosomes, while the box indicates the respective genotype of the cytoplasm. The haploid inducer line was obtained in a Col-0 genotypic background (green). Note that the haploids (monoploids) and doubled haploids (diploids) retain the cytoplasm of the haploid inducer line, while the F_2_ population retains the cytoplasm of the original F_1_.

**Figure 2 genes-14-01161-f002:**
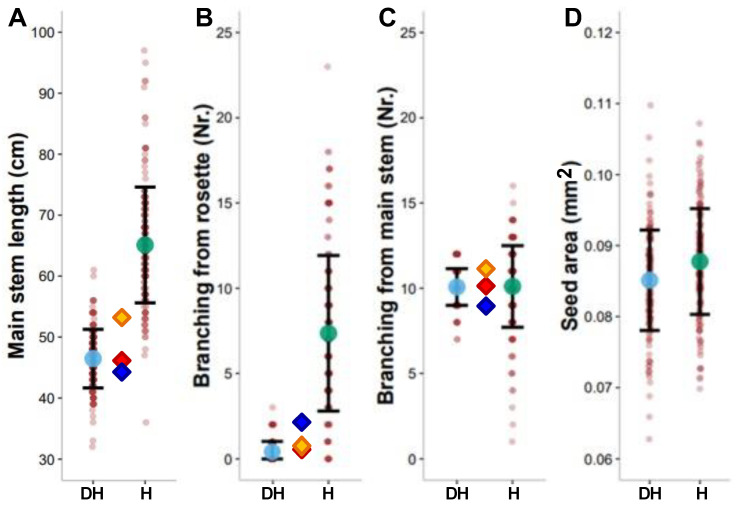
Distribution of morphological trait values in monoploid and diploid Arabidopsis plants. The mean value of diploids (DH) and monoploids (H) is indicated with transparent blue and green dots, respectively. The shaded dots depict the value of individual monoploids and the line average of five replicates for each diploid genotype, respectively. Diamonds represent mean trait values of the parental lines Ge-0 (red) and T540 (blue) and their F_1_ hybrid (orange). Error bars indicate the standard deviation of the mean.

**Figure 3 genes-14-01161-f003:**
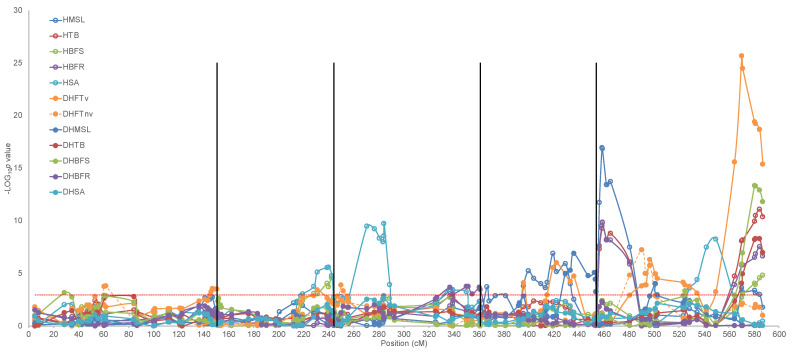
QTL plots of morphological traits mapped in a haploid and diploid segregating population. Traits were mapped in a single multi-trait CIM analysis. Chromosomes are separated by vertical black lines while map positions are indicated on a continuous scale. The horizontal dotted red line indicates the significance threshold (-LOG_10_p 2.964). Open and closed symbols represent traits measured in the haploid and diploid population, respectively. Solid lines represent traits measured in vernalised plants while the dashed line represents flowering time measured without vernalisation. MSL, main stem length; TB, total branching; BFS, branching from stem; BFR, branching from rosette; SA, seed area; FT, flowering time. H, haploid; DH, doubled haploid; v, vernalised; nv, non-vernalised.

**Table 1 genes-14-01161-t001:** Flowering time of parental accessions and their derived populations with and without vernalisation.

Trait	Genotype	Mean (*n*)	s.d.	Min	Max	Cv (%)
FTv	Ge-0	14.2 (29)	1.17	13	17	8.2
	T540	19.1 (22)	1.50	15	22	7.8
	F1	17.1 (18)	0.80	16	18	4.7
	F2	15.4 (172)	1.34	13	20	8.7
	DH	15.9 (171)	1.57	12	22	9.8
FTnv	Ge-0	86.5 (8)	16.27	64	107	18.8
	T540	101.8 (4)	14.08	89	121	13.8
	F2	87.6 (180)	16.41	60	130	18.7
	DH	88.4 (163)	12.09	63	123	13.7

FTv, flowering time after vernalisation (days after transfer); FTnv, flowering time without vernalisation (days after sowing); s.d., standard deviation; Cv, coefficient of variation.

**Table 2 genes-14-01161-t002:** QTL detection for flowering time with and without vernalisation in a DH and F_2_ population.

Population	Chromosome	Position (cM)	−LOG_10_(*p*)	Trait	*p*-Value	Effect-Size	s.e.	%EV	Type
DH	I	70.8 (5.2–150.6)	4.5	**FT_nv**	**<0.001**	**0.31**	**0.08**	**9.8**	-
				**FT_v**	**0.005**	**0.16**	**0.06**	**2.4**	-
	I	149.5 (5.2–150.6)	4.4	FT_nv	0.391	0.06	0.07	0.3	-
				**FT_v**	**<0.001**	**0.21**	**0.05**	**4.4**	-
	II	80.4 (0.9–93.5)	2.3	FT_nv	0.120	−0.11	0.07	1.1	-
				**FT_v**	**0.002**	**−0.15**	**0.05**	**2.3**	-
	III	5.9 (2.9–117.2)	4.9	**FT_nv**	**<0.001**	**0.26**	**0.07**	**6.8**	-
				**FT_v**	**<0.001**	**0.16**	**0.05**	**2.7**	-
	IV	61 (5.3–92.8)	10.9	FT_nv	0.653	−0.03	0.07	0.1	-
				**FT_v**	**<0.001**	**0.33**	**0.05**	**10.9**	-
	V	61.2 (2.4–133.1)	9.7	**FT_nv**	**<0.001**	**−0.43**	**0.07**	**18.7**	-
				**FT_v**	**<0.001**	**−0.21**	**0.05**	**4.3**	-
	V	121.7 (75.4–133.1)	38.7	**FT_nv**	**0.004**	**0.19**	**0.07**	**3.7**	-
				**FT_v**	**<0.001**	**0.65**	**0.05**	**42.6**	-
F2	IV	5.3 (5.3–92.8)	3.2	**FT_v**	**-**	**0.26**	**0.19**	**2.0**	**Additive**
					**-**	**0.98**	**0.27**	**-**	**Dominance**
	V	126.4 (109.5–133.1)	5.7	**FT_v**	**-**	**1.21**	**0.23**	**44.0**	**Additive**
					-	-	-	-	Dominance

QTL positions are presented in cM with support intervals between brackets. −LOG_10_(*p*) indicates the significance of the QTL for the combined treatments, while the *p*-value provides the specific *p*-value for each treatment. FT_nv, flowering time without vernalisation (days after sowing); FT_v, flowering time after vernalisation (days after transfer). Effect-size is given as the normalized additive effect of the QTL, where positive values indicate a positive effect of the T540 allele and negative values indicate a positive effect of the Ge-0 allele; s.e. is the standard error of the mean effect; %EV is the explained variance according to a mixed model. For the F_2_ population dominance effects could be calculated, which are indicated as Type. Significant QTL effects on flowering with or without vernalisation are indicated in bold.

**Table 3 genes-14-01161-t003:** QTL detection for phenotypic variation in a monoploid and diploid segregating mapping population.

Trait	Chromosome	Position (cM)	-LOG_10_(p)	Population	*p*-Value	Effect-Size	s.e.	%EV
MSL	III	67.4 (2.9–117.2)	4.1	Diploid (DH)	0.234	0.10	0.08	0.9
				Monoploid	<0.001	−0.24	0.07	5.9
	IV	5.3 (5.3–92.8)	3.7	Diploid (DH)	0.033	−0.15	0.07	2.3
				Monoploid	<0.001	−0.24	0.06	5.8
	IV	57.9 (5.3–92.8)	3.3	Diploid (DH)	0.030	−0.18	0.08	3.2
				Monoploid	<0.001	−0.26	0.07	6.9
	IV	88.8 (5.3–92.8)	4.0	Diploid (DH)	<0.001	−0.36	0.08	12.6
				Monoploid	0.099	−0.12	0.07	1.3
	V	4.7 (2.4–133.1)	17.5	Diploid (DH)	0.698	0.03	0.07	0.1
				Monoploid	<0.001	−0.49	0.06	23.5
BFR	III	102.5 (2.9–117.2)	3.5	Diploid (DH)	0.001	0.26	0.08	7
				Monoploid	0.009	0.18	0.07	3.2
	V	4.7 (2.4–133.1)	9.3	Diploid (DH)	0.006	0.20	0.08	4.2
				Monoploid	<0.001	0.41	0.07	17
	V	130.5 (2.4–133.1)	6.6	Diploid (DH)	0.642	−0.03	0.07	0.1
				Monoploid	<0.001	−0.35	0.07	12.5
BFS	II	84.3 (0.9–93.5)	4.1	Diploid (DH)	0.010	−0.17	0.07	2.9
				Monoploid	<0.001	−0.30	0.08	8.7
	V	71 (2.4–133.1)	3.7	Diploid (DH)	<0.001	−0.23	0.06	5.3
				Monoploid	0.204	0.09	0.07	0.9
	V	126.4 (2.4–133.1)	14.4	Diploid (DH)	<0.001	−0.48	0.06	23.1
				Monoploid	<0.001	−0.30	0.07	8.8
SA	II	84.3 (0.9–93.5)	6.0	Diploid (DH)	0.004	0.23	0.08	5.3
				Monoploid	<0.001	0.31	0.06	9.8
	III	21.6 (2.9–117.2)	8.1	Diploid (DH)	0.043	−0.17	0.08	2.9
				Monoploid	<0.001	−0.42	0.07	17.4
	III	62.7 (2.9–117.2)	2.2	Diploid (DH)	0.224	−0.11	0.09	1.3
				Monoploid	0.002	−0.24	0.08	5.5
	V	91.8 (2.4–133.1)	8.0	Diploid (DH)	0.006	−0.21	0.08	4.5
				Monoploid	<0.001	−0.37	0.06	13.4

Positions of detected QTLs are shown in cM with support intervals between brackets. -LOG_10_(p) indicates the significance of the QTL for the combined ploidy levels, while the *p*-value specifies the significance for each level. Effect-size is given as the normalized additive effect of the QTL, where positive values indicate a positive effect of the T540 allele; s.e. is the standard error of the mean effect; %EV is the explained variance according to a mixed model. QTLs with a significant *p*-value (<0.05) are indicated in solid text, while non-significant QTLs for a specific ploidy level are noted in grey. MSL, main stem length (cm); BFR, branching from rosette (nr.); BFS, branching from stem (nr.); SA, seed area (mm^2^).

## Data Availability

The data presented in this study are available in Appendix A.

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
