# Peer review of "Genetic Mapping of Genotype-by-Ploidy Effects in Arabidopsis thaliana"

_genes, 2023, doi:10.3390/genes14061161_

Round 1

Reviewer 1 Report

This study provides valuable insights into the genetic factors underlying ploidy-dependent phenotypic variation in Arabidopsis thaliana. The study was well-organized and the results are solid, and the writing is clear and fluent. 

The authors utilized a haploid-inducer line to develop recombinant haploid and diploid offspring, enabling a comparison of phenotypes between these two groups. This approach allowed for the detection of ploidy-specific quantitative trait loci (QTLs), and a multi-trait analysis revealed pleiotropic effects for several of these QTLs. The study provides compelling evidence that genetic variation between different Arabidopsis accessions causes dissimilarities in phenotypic responses to altered ploidy levels. Moreover, the identification of a major vernalisation-specific QTL for variation in flowering time counters the historical bias of research in early flowering accessions.

One minor point for improvement is that the resolution for figure 2 is low. A high-resolution figure would be more beneficial. Additionally, the colored points in figure 2 are somewhat busy, and it may be helpful if the authors label the names of each group on the x-axis to enhance clarity. 

Overall, this study is solid and highlights the importance of considering ploidy level and genetic variation in the analysis of phenotypic variation.

Author Response

We have adjusted figure 2 according to the reviewer's comments.

Reviewer 2 Report

The manuscript "Genetic mapping of genotype-by-ploidy effects in Arabidopsis thaliana" addresses the interesting topic of ploidy-dependent phenotypic response from the genetic point of view. This is indeed an underexplored research field for which this kind of study is very welcome.

The authors identify an interesting set of QTLs and elaborate on the statistical significance of QTLs across different ploidy levels. As a result, pleiotropic effects associated with the interaction between ploidy and genotype are identified. Furthermore, the results suggest that genetic variation's effect seems stronger in monoploids than in diploids.

Producing haploids and double haploids is a good strategy for comparing identical genomic configurations at two different ploidy levels. This is a straightforward but, at the same time, robust logical experimental design.

Unfortunately, a significant flaw in this study undermines the reliability of all presented results. The authors do not mention any flow cytometry analysis performed on the studied plant material. So how is it possible to talk about haploids and diploids if no flow cytometry measurements were performed?

Without this kind of information, the study is missing an essential piece, and this is a pity because the manuscript is very interesting and addresses a crucial biological question in a simple and logical way.

The author must address this point by showing flow cytometry measurements for both monoploids and diploids (at least measurements from 10 individuals for each group).

While diploid material is easy to germinate and analyze, I wonder if the authors still have putatively-haploid material available.

Without this flaw, I would suggest acceptance with minor revisions (listed below), but if this point is not fixed, it will be impossible to publish this work. Therefore, as a result of this evaluation, I suggest major revisions.

The English level of the manuscript is appropriate, and all parts are well-written and understandable.

Please check some minor points listed below.

Line 2 – Arabidopsis thaliana must be written in italics, even if it is in the title.

Line 180 – Why the authors used the flower heads for DNA extraction? Foliar tissue would have been less destructive and allowed to continue the phenotyping of that plant. Please explain why you needed to extract the DNA from floral heads.

Line 340 – The authors observed that 60% of the monoploids grew taller than the tallest diploid. How can they be sure that the remaining 40% were also monoploids? This connects to my point that flow cytometry is crucial for this type of study and is missing here.

Figure 3 – Please specify in the legend the meaning of the following letters:

H = haploid

D = diploid

v = vernalized

nv = non-vernalized

Author Response

Minor points:

Line 2 – Arabidopsis thaliana must be written in italics, even if it is in the title.

It is and was written in italics.

Line 180 – Why the authors used the flower heads for DNA extraction? Foliar tissue would have been less destructive and allowed to continue the phenotyping of that plant. Please explain why you needed to extract the DNA from floral heads.

Flower head sampling is the most efficient and reliable protocol for DNA extraction. We agree with the reviewer that taking a leave would be less destructive but does not guarantee the absence of effects completely. We chose to phenotype undisturbed plants when possible and only included traits of genotyped plants before flower head clipping.

Line 340 – The authors observed that 60% of the monoploids grew taller than the tallest diploid. How can they be sure that the remaining 40% were also monoploids? This connects to my point that flow cytometry is crucial for this type of study and is missing here.

See our response to the major comment.

Figure 3 – Please specify in the legend the meaning of the following letters:

H = haploid

D = diploid

v = vernalized

nv = non-vernalized

We have adjusted the legend accordingly.

Major point:

So how is it possible to talk about haploids and diploids if no flow cytometry measurements were performed?

This is a valid point raised by the reviewer. However, we chose not to analyse the phenotyped plants by flow cytometry for the following reasons:

  1. Flow cytometry requires quite a bit of plant material and sampling would disturb the development of plants too much to acquire accurate measurements. Since haploids are unique, ploidy measurements, genotyping and phenotyping needs to be performed on the same plant. We chose to prioritize on the phenotyping and rely on other assessments for ploidy and genotyping.
  2. Haploid plants can easily be distinguished morphologically from aneuploids and diploids in an early developmental stage. Furthermore, haploid plants are sterile while diploids are not. And finally, doubled haploids can only result from haploids in contrast to aneuploids.
  3. Alternative genotypes (e.g. diploids resulting from failing genome elimination) would be identified by genotyping and including such lines would dramatically confound genetic map construction and QTL mapping. We have not experienced either of these phenomena.
  4. Genome elimination has been extensively investigated cytogenetically (Ravi and Chan, 2010) and has been proven to be a reliable method to generate haploids and doubled haploids in the mentioned study and many papers following this initial study.

For those reasons we are quite confident that our assumptions about the ploidy level of the investigated resources are correct and we, therefore, saw no reason to validate this with flow cytometry. However, to meet this reviewer's concerns, we have added a sentence stating that haploids can easily be distinguished from diploids and aneuploids.

Reviewer 3 Report

The manuscript related to QTL mapping related to genotype-ploidy in A. thaliana has identified several genomic regions that can be considered as potential regions for genetic studies. The paper also written well and the English language is fluent.

 Additional comments are listed here

# comment-1

 In the introduction part only advantages of DH versus inbred line are described. Please add also disadvantages of DH lines that might of interest to reader

Author Response

We have adjusted the introduction according to the reviewer's comment.

Round 2

Reviewer 2 Report

The authors of the manuscript "Genetic mapping of genotype-by-ploidy effects in Arabidopsis thaliana". Have addressed many points raised during the first round of review.

Nevertheless, the only critical point was about the validation of ploidy levels through flow cytometry, and it was not addressed.

The authors bring four points as justification for not performing this work.

  1. Flow cytometry requires quite a bit of plant material and sampling would disturb the development of plants too much to acquire accurate measurements. Since haploids are unique, ploidy measurements, genotyping and phenotyping needs to be performed on the same plant. We chose to prioritize on the phenotyping and rely on other assessments for ploidy and genotyping.
  2. Haploid plants can easily be distinguished morphologically from aneuploids and diploids in an early developmental stage. Furthermore, haploid plants are sterile while diploids are not. And finally, doubled haploids can only result from haploids in contrast to aneuploids.
  3. Alternative genotypes (e.g. diploids resulting from failing genome elimination) would be identified by genotyping and including such lines would dramatically confound genetic map construction and QTL mapping. We have not experienced either of these phenomena.

Reply to point 1:

Flow cytometry in Arabidopsis can be done with a fraction of a leaf. I disagree that “quite a bit” of plant material is needed.

Reply to point 2:

The genome elimination system can create haploids but also aneuploids. The phenotypic difference between haploids, aneuploids, or diploids is not always obvious. Furthermore, not even a photo of a plant is included in support of this point.

Reply to point 3:

Genotyping gives you information on the sequence of the genome at different loci, but it is not a good reason to skip the ploidy measurements because the different ploidy levels discussed in the manuscript are the basis of most of the work during this study.

Reply to point 4:

Genome elimination is indeed reliable, but this does not justify the total skipping of flow cytometry. This is a pretty simple analysis that should be presented to the reader before starting to elaborate on any result connected to ploidy.

Author Response

We fully understand the concerns of this reviewer and yes, perhaps we should have carried out flow cytometry to convince the sceptic reader. However, we have not done so for reasons explained earlier. We further like to emphasize that we are not reluctant or stubbornly refusing to do these analyses but it is simply no longer possible since haploid plants cannot be maintained and hence this material is no longer available.

We would like to stress once more that none of the results described in the manuscript would have been obtained if ploidy levels would have been misinterpreted. E.g. doubled haploids could not have been generated from anything else but true haploids and genotyping easily discriminates heterozygotes form homozygotes (i.e. F2s vs DHs). More importantly, the construction of the genetic map and the subsequent QTL mapping would be dramatically compromised if the investigated populations were derived in any other way than assumed, leading to nothing else but nonsense results.

Nonetheless, to meet the reviewer’s concerns we have now included a more elaborated explanation of our choices so the reader can decide for itself whether this is sufficiently convincing to support our results.

We regret that our study has caused doubt on the reliability of our findings but at this point we see no other options than to either follow our reasoning and accept the manuscript as is or to reject publication on grounds raised by the reviewer.

Round 3

Reviewer 2 Report

In light of the critical point regarding the ploidy measurement that has not been or cannot be addressed, I cannot recommend the manuscript for publication.

Author Response

See reply to editors.